# Promoting Rural Regeneration and Sustainable Farming near Cities Thanks to Facilitating Operators in France? The Case of the Versailles Plain's Association Governance Model

Camille Robert-Boeuf 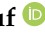

Centre National de la Recherche Scientifique (CNRS), UMR Ladyss, 92000 Nanterre, France;
c.robertboeuf@outlook.com; Tel.: +33-768036298

**Abstract:** Food and agricultural systems in rural areas close to cities have been the subject of much academic research, revealing difficulties due to the proximity of cities, land pressure, and complex governance between cities and rural areas. This article aims to analyze the case study of the Versailles Plain Association (VPA), which proposes an original form of facilitation that contributes to the effectiveness of territorial governance in a rural area close to Paris. It is based on a qualitative method, gathering 52 interviews and heritage audits that were carried out in the framework of the European project H2020 Ruralization. The analysis shows that the VPA is a facilitating operator that brings together stakeholders from both the agricultural and urban worlds, allowing collective projects around a territorial identity. This territorial identity promotes agriculture and rural lifestyles, which become positive embodiments of local development. This facilitating operator thus offers an alternative to urbanization and produces forms of ruralization processes.

**Keywords:** peri-urban agriculture; sustainable farming; rural regeneration; facilitation; territorial governance; ruralization

## 1. Introduction

Urbanization has impacted rural areas in several ways in developed and developing countries: urban sprawl and land pressure have led to the artificialization of agricultural and natural areas and to an increase in land pressure [1–3]. The increase in urban populations in rural areas close to cities has changed lifestyles and densified built-up areas, causing tensions between rural populations and newcomers [4,5]. In addition, urbanization challenges rural ways of living, and rural territories are becoming increasingly dependent on towns and cities [2,6].

In this context, rural-urban relationships are increasingly complex and are facing many challenges in Europe: demographic and economic crises, global warming, and agribusiness crises [7,8]. In this context, the H2020 Ruralization project aims to show processes of rural regeneration that will help define the new contours of European ruralization, i.e., a synergetic process that promotes new rural opportunities and development to respond to urbanization. The concept of ruralization has been used very little in academic literature [9–11]. Often in the post-Soviet context ruralization refers to the rural areas' crises in Eastern Europe with the disintegration of the countryside (marked by a decrease in farming populations and agricultural decline) [12,13]. Ruralization as a positive response to urbanization is therefore original [14], and in this paper, we wish to provide some elements to define and evaluate the potential of this process in France.

In rural areas close to cities and peri-urban contexts, urbanization has a strong impact on the farming sector and farmland, leading to the construction of new governance, networks, and policies at a local level [15,16]. These cooperation models often create power relations between urban decision-making authorities and rural populations who are directly

confronted with the consequences of urbanization [15,17]. This raises questions regarding the empowerment of rural populations, especially farmers, near cities [18,19].

Furthermore, international academic literature shows that promising initiatives often seem to come from agricultural areas close to cities and from local authorities or civil societies [19,20]. However, local food supplies have difficulties coordinating and organizing common projects with very different stakeholders (such as farmers, local officials, and NGOs) [21,22]. New entrants into farming are confronted with particular issues (they have less access to land, difficulty getting loans, etc.) [23,24], mostly in peri-urban areas [25,26]. Studies on peri-urban areas and peri-urban agriculture have been conducted since a long time in the European context, particularly in France [27,28]. Despite urbanization, land pressure, and socio-economic difficulties, they highlighted positive relations between city and countryside, interdependency between city dwellers and farmers [25,26,29], and the positive impact of peri-urban farming on the transition towards sustainable agriculture [30].

In France, rural areas are deeply impacted by urbanization because 55,000 hectares of agricultural and natural land are artificialized every year [31], and the number of farmers has been drastically decreasing for several years [32]. At the same time, agricultural policies are increasingly being implemented at the territorial level and in a bottom-up manner [33,34]. Since 2014, the state has been funding a new policy to promote local food systems called Territorial Food Projects that are driven by local authorities (agglomerations and communities of communes) or NGOs [35,36]. For several years now, thanks to decentralization, metropolises and urban regions have also been working to establish agricultural policies, which aim to preserve agricultural land and maintain farms for a short food supply [20].

Thus, innovative projects for agricultural sustainability at the local scale tend to develop in rural areas near cities [26,35,37]. However, the "local" is often hard to define and can be understood very differently depending on different studies [38], and governance strategies are confronted by the diversity of stakeholders with divergent interests working in formal and informal networks [15,39]. Using the Ruralization theoretical framework and based on the Versailles Plain Association case study, this paper aims to assess the role of facilitation bodies in maintaining agricultural areas close to cities: what kind of facilitating operator can promote rural regeneration in peri-urban areas? How does a facilitating operator contribute to a better local and rural-urban governance model as well as to empowerment processes for local stakeholders? To what extent do facilitating bodies promote sustainable agriculture and a positive territorial identity in peri-urban contexts?

## 2. Materials and Methods

### 2.1. Theoretical Framework

For this paper, we first used the framework of the European project H2020 Ruralization. In this framework, rural regeneration is understood as a multi-dimensional process that goes beyond just reversing rural decline. This implies a process of transition and a more positive and resilient reinvention or revival [40,41]. Rural regeneration is also a pathway promoting the renewal of farmers (successors or new entrants into farming) and sustainable agriculture through multidimensional practices in rural areas [42,43]. This pathway is complex, especially in peri-urban areas, where land and demographic pressures strongly impact farming and food activities.

Rural regeneration is also deeply linked to generational renewal. Indeed, the sharp decline in the number of farmers in all European countries raises questions about the aspirations of new generations [44,45]. Moreover, at the local level, a positive impact on the economy of new populations has been shown [46,47]. Therefore, we examined the role of new populations (whether they are new arrivals in rural areas or new entrants into farming or successors) in the regeneration of the rural areas studied.

From this perspective, our analysis is integrated in a research field at the crossroads of geography and sociology on the construction and representations of local territories. Participating in the scientific literature on food systems [38,48,49] and territory construc-



tion [50], we placed the concepts of territory and network at the core of our study. In this paper, we are very much in line with the territorial scheme of Lamine et al. [51,52] and Felici and Mazzocchi [38], and affirm that an innovative pathway at a territorial level relies on the diversity of stakeholders, and therefore, on their network and governance. Therefore, place-based analysis is a structuring perspective, as local, regional, and national geographical contexts are central to understanding the different pathways and dynamics of rural regeneration [53,54].

To better assess territorial governance and networks of the Versailles Plain, we focused on the facilitating process of the Versailles Plain Association (VPA), contributing to academic studies on local and territorial governance [55–57], and on facilitation frames [39,58]. Unlike previous studies showing that facilitation processes coming from non-governmental actors illustrate a greater "responsabilisation" of these actors and a withdrawal of the state [59,60], we hypothesize that a facilitation process can bring urban and rural stakeholders (farmers, urban elected officials, and NGOs) together and produce forms of empowerment [18,61,62] for the regeneration of rural territories close to cities.

### 2.2. Presentation of the Case Study

The Versailles Plain is located to the west of the richest French region, in the Yvelines Department of the Île-de-France region.

Even though Paris is at the center of this territory, 50% of Île-de-France is occupied by agricultural land (see Table 1). In the 19th century, agricultural land was in the market garden belt of the capital city [63]. In the second half of the twentieth century, the metropolis spread over Paris's borders, including peri-urban spaces and urbanizing agricultural land [64]. Today, it is mostly occupied by cereal farms; cereals and oilseeds cover nearly 80% of the regional Utilized Agricultural Area (UAA). However, since the 2000s, vegetable farms have been developing again and, for example, the region ranks 5th nationally in terms of producing potatoes, eggplants, and radishes [65,66].

**Table 1.** Agricultural characteristics of Île-de-France Region and Yvelines Department in 2020 (source: Agreste and Regional Agricultural Chamber).

|  | Île-de-France Region | Yvelines |
|---|---|---|
| Total UAA (ha) | 569,000 | 89,291 |
| Average area per farm (ha) | 127 | 94 |
| Farms number | 4425 | 807 |
| Number of organic farms | 600 | 152 |
| Cereal farms UAA (ha) | 505,020 | 54,480 |
| UAA of Market gardeners, arborists, and winegrowers (ha) | 10,361 | 1998 |
| UAA of Horticulturists (ha) | 1118 | 326 |
| Number of cattle | 27,463 | 7273 |
| Number of pigs | 7175 | 216 |
| Number of goats | 2087 | 962 |
| Number of sheep | 10,360 | 2192 |

The Yvelines Department has concentrated agricultural land and large forests, which represent 34% of the area. It is a relatively low-density area (629 inhabitants per km$^2$) compared to the Île-de-France region (1017 inhabitants per km$^2$) [67]. It is a rich territory with a much lower poverty rate (9.7% compared to 15.6% in the Île-de-France Region) and an unemployment rate of 10.4%, which is below the regional average (12.5%) [67]. Nevertheless, the Yvelines department also has concentrated municipalities with significant social gaps in the south of Versailles Plain.

In this geographical context, the Versailles Plain is an ancient cereal plain of 23,000 hectares surrounding the Versailles Castle. This plain borders the Gally Rivulet, which flows from east to west and is surrounded by two plateaus to the north and south, where important cities are located (see Figure 1).

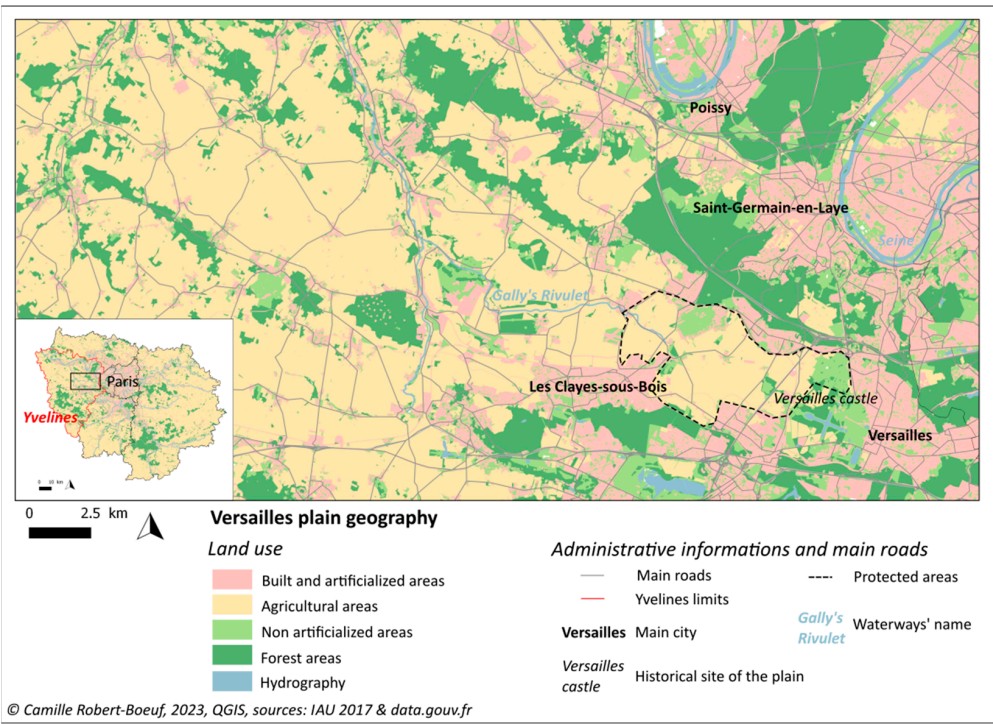

**Figure 1.** Local context of the Versailles Plain.

The Versailles Plain has relatively small farms but illustrates the traditional agricultural landscape of the Île-de-France Region: 83% of the UAA is in cereals and oil-protein crops (including 50% in soft winter wheat) [66]. However, this plain, coherent with its geography, did not have its own administrative structure: it includes 25 municipalities and five commune communities within the Yvelines Department. The lack of a structural body has been a weakness in the context of increasing urbanization and densification of the region. Currently, the urban area of Paris largely exceeds the limits of the region. In recent years, the "Grand Paris" project and the planning of the 2024 Olympic Games accelerated the urbanization process, impacting the Versailles plain [68,69].

This geographical context caused the mobilization of several historic entrepreneurial farmers who wanted to go further than their individual projects of local food supply chains and carry farming issues in local political bodies. For many farmers, the future was full of uncertainties because of several international, national, and local trends: the future of the Common Agricultural Policy (CAP), low visibility on land, weakened links with urban areas, and increasing challenges for the intensive agricultural model. In addition to exogenous urban pressures, there was the endogenous problem of agricultural activity's capacity to maintain itself in its areas. In 2000, the state classified 2600 ha of farmland located in the immediate vicinity of the Versailles Castle, but it created tensions with local farmers because protected land has strict regulations, increasing costs for maintenance of farms (e.g., farmers setting-up on protected sites must use local or natural materials for farmhouse renovations). Moreover, these protected areas covered only a small part of the plain and shifted urbanization further to the west of the plain. To respond to these challenges, farmers created the Versailles Plain Association in 2004.

The Versailles Plain context appeared to be "promising" because it had concentrated on innovative practices for more than twenty years, which made it possible to have a long-term analysis of the agricultural sector and its territorial dynamics in a complex peri-urban context.

*2.3. Data Collection*

The data collection methodology was qualitative and conducted in two parts: (i) survey with 52 semi-structured interviews collected (see Table 2); (ii) collaborative research with the Versailles Plain Association.

**Table 2.** Type of stakeholders interviewed.

| Fieldwork Stages | Type of Stakeholders | Number of Interviewees | Female | Male |
|---|---|---|---|---|
| Online survey (2020) | Farmers | 5 | 3 | 2 |
| | Food Artisans | 2 | - | 2 |
| | VPA leaders | 3 | 2 | 1 |
| | Elected officials and institutions * | 8 | 3 | 5 |
| | NGOs | 11 | 3 | 8 |
| | Others | 2 | 2 | - |
| Fieldwork in Versailles plain (2021) | Farmers | 7 | 2 | 5 |
| | Food Artisans | 4 | - | 4 |
| | VPA leaders | 2 | 1 | 1 |
| | Elected officials and institutions | 5 | 3 | 2 |
| | NGOs | 3 | 1 | 2 |
| | Total | 52 | 20 | 32 |

* We interviewed local elected officials, mayors, and state institution officials who have an impact on the plain.

Our team conducted semi-structured interviews in two stages in 2020 and 2021:

In April and May 2020, because of the COVID pandemic, we organized an online study with a group of students, that produced a "patrimonial audit" [70] of the plain with 31 stakeholders to identify the problems and the actors' relationship, as well as to establish a diagnosis of the territory and draw up a prospective analysis. The patrimonial audit methodology was based on a 4-part interview: (i) identification of stakeholders and issues; (ii) diagnostic of the actual situation; (iii) prospective strategies and scenarios; and (iv) action proposals. This survey grid made it possible to draw up a dynamic portrait of the Versailles Plain with the interviewees.

Then, in March and April 2021, we conducted fieldwork on the plain and conducted 21 interviews to analyze the governance of the VPA and its impact on rural and farming regeneration (see Figure S1).

In addition, a collaborative research approach was added to the VPA. We conducted participant observations at meetings of the association. We also organized eight meetings between 2020 and 2023 with VPA members to (i) present our survey approach and results and (ii) discuss the survey's results and integrate their remarks.

In 2022, our work was integrated into the VPA project to apply for European Leader project funding. The results of our fieldwork participated in VPA members' debates on the current difficulties, territorial evolution, and future of the plain for this appliance. For this perspective, we participated and co-animated two workshops on the European Leader project appliance with VPA.

This collaborative research has allowed us to meet with stakeholders other than those who were interviewed. Thanks to observation situations, we were able to analyze actors' discourses and relationships and better understand how the VPA network formally and informally works.

## 3. Results and Discussions

*3.1. A facilitating Operator to Link Rural and Urban Stakeholders*

To resolve farming issues and defend agricultural land from urbanization, the Versailles Plain's Association built a facilitation process that brings together urban and rural stakeholders. Thus, the VPA became a facilitating operator which came from the initiatives

of farmers and not from state actors, as has been analyzed in other studies on urban and territorial governance [15,20,57].

Since the creation of the VPA in 2004, the association is based on three different colleges: farmers' college, elected officials' college, and civil society's college. Each of these three groups has nine members, and together they form the Heritage Council. This original structure, with different members, promotes the rich diversity of the actors involved. This horizontal organization was designed to put all actors on the same level and especially to put farmers on the same footing as elected local officials. The first strength of VPA governance is the central place of farmers in the network: they are at the core of the facilitating body; they were involved at the very beginning of the VPA and now represent around one-third of the farmers working in the plain. In new agricultural policies in thr peri-urban context, studies have shown that farmers were often on the margins of the decision-making process [26,71]. The VPA legitimizes farmers as important stakeholders in urban planning and urban governance. They bring their claims to the attention of local and regional authorities by promoting their role in regional planning (such as the production of reports on the impact of farmers on environmental protection or territorial diagnoses), by disseminating farmers' arguments and by making their actions visible (territorial marketing in the plain, lobbying on social networks).

The second strength of the VPA is its capacity to bring together different stakeholders who usually do not work together and to link rural and urban worlds. First, they bring together several types of farmers: conventional farmers and organic farmers with food start-ups and small-scale market gardeners; and cereal farmers with large farms to successors and new entrants to farming. The VPA proposes training and workshops for all farmers and creates a common space to exchange and produce common revendications. As the Versailles Plain has very diverse agriculture (see Table 3), the VPA represents this diversity.

**Table 3.** Versailles plain's farming activities in 2018.

| Total of Farmers | Cereal FARMERS | Market Gardeners and Fruit Growers | Equestrian Centers and Horse Breeders | Horticulturalists | Beekeepers | Chicken Breeding |
|---|---|---|---|---|---|---|
| 124 | 76 | 23 | 12 | 7 | 4 | 2 |

© Author, 2023, sources: Insee, 2018 & Study of VPA, 2018 (https://www.plainedeversailles.fr/les-etudes-de-lappvpa, accessed on 10 September 2022).

The VPA also federates urban and rural elected officials around farming issues and facilitates relationships with agricultural institutions, which have traditionally communicated very little with urban institutions. Indeed, French urban and farming institutions are working separately, but as food policies are increasingly driven by urban actors, agricultural institutions and farmers tend to be less integrated [20,72]. The VPA network brings together various elected officials from rural and urban municipalities to make them aware of agricultural issues and mobilize them to setup farmers' projects or to protect agricultural land. Thus, its network goes beyond the administrative boundaries that divide the plain and promotes the Versailles Plain not as an administrative entity, but as a geographical and historical area.

The VPA also facilitates relationships and exchanges between agricultural institutions and political officials. As it concentrates on a large group of farmers, the association has extensive knowledge about the agricultural institutions' functioning and can communicate relatively well with them. The association collaborates with the SAFER (Land Development and Rural Establishment Organization) and the Agricultural Chamber when there is land to sell in the Versailles Plain or when new farmers (new entrants or successors) want to setup in the plain. Therefore, the association is a concrete connection between stakeholders and institutions; it helps them exchange, and disseminates its knowledge regarding institutions, setting-up procedures, and funding-support applications for farming activities.

The effectiveness of the VPA network is illustrated, for example, in an urban municipality east of the plain (near Versailles) with a population of 10,975, where local elected

officials and the VPA have actively helped setup a market gardener. The market gardener wanted to buy two hectares in the municipality and had gone to the town hall for support and to make himself known to the population (he wanted to sell directly from the farm and was looking to contact potential customers). However, in France, the SAFER has legal control over farmland selling [73], and it gave a negative opinion to the market gardener project. This position was caused by the viability of the traditional economic system of the Versailles plain, where most farmers are usually cereal farmers and not market gardeners. Therefore, the SAFER considered that its project was not sustainable in the territory. The VPA intervened to bring together the elected officials of the municipality, the farmer, and the SAFER representatives. This led, after the farmer had submitted a second setting-up project, to the purchase of the land and the successful setup of the farmer.

> *"My partner set up in 2018 on 5 hectares and I joined him in 2021. We supply the local AMAP [Association for the Maintenance of Peasant Agriculture] twice a week, 100 baskets a week and that represents 150 families. [ . . . ] We are very well supported by the municipality and the inter-municipalities. We've had meetings with the local elected official, who is really excited about our set-up here, because it's for the municipality and its inhabitants that we're setting all this up. [ . . . ] At the beginning of the project, we immediately went to see the municipality and we were in contact with the VPA. The VPA communicates quite a lot about the plain, it's good, they support us well in our projects."*
> *New entrants into farming, Versailles plain*

The VPA network successfully connects farmers, elected officials, and local institutions with civil society and NGOs (see Figure 2). It creates a proper space for NGOs to debate, where NGOs in conflicts—for example, environmental and hunting NGOs—can try to find a common point of view regarding the Versailles plain conservation. Moreover, one of the strengths of the VPA is that it connects not only different types of actors but also different generations of actors. NGOs and farmers' groups connected with the VPA concentrate on young and older stakeholders who hardly communicate.

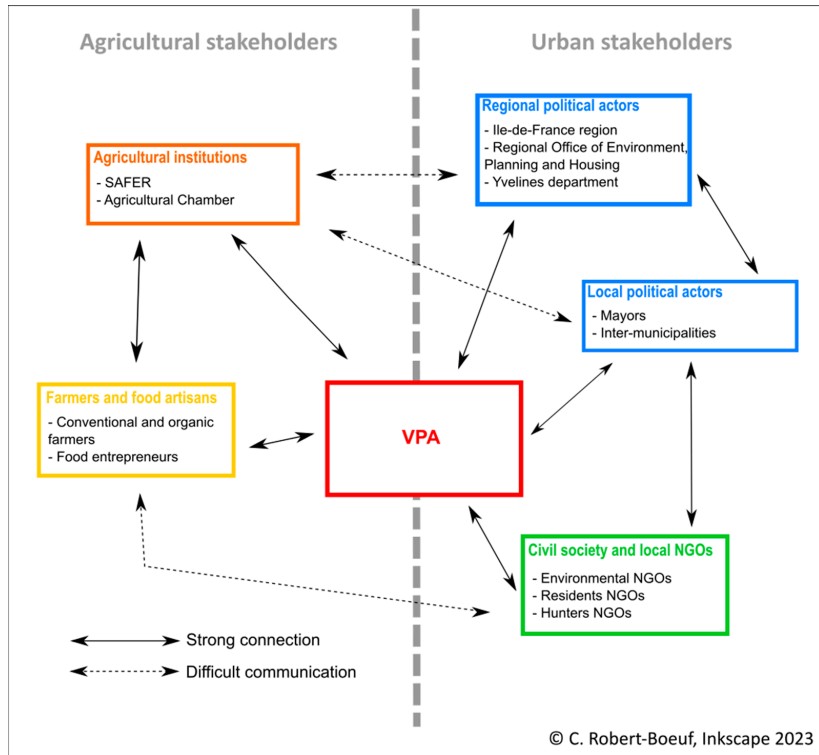

**Figure 2.** VPA network.

The VPA is, therefore, a physical platform (since it has premises in the center of the plain where it can welcome people), but also a network that allows exchange and dialogue between all the stakeholders in the plain. This case study is in line with Lamine's analysis of the reconfiguration of the food systems' local governance, but shows, contrary to Lamine's work, that a balanced governance between political actors and farmers is possible [51,52]. Indeed, the VPA has succeeded in creating a "space of governance" [38] that does not depend only on political actors, because its associative nature and organization with three colleges allows all actors to participate as equals.

However, the VPA needs political support, which is variable and questioned at each election, making it difficult to establish long-term actions or projects for the VPA. This lack of long-term vision is reinforced by the VPA's lack of regular financial resources. Faced with these difficulties, we agree with the analyses of Lamine and Felici and Mazzocchi [38,52] on the difficulty of maintaining local governance in the long term outside of political and administrative structures. The VPA network needs constant reconfigurations and new arrangements between stakeholders, and thus, continuous renewal of active actors.

### 3.2. The VPA an Operator to Facilitate Local Food Supply and Economic Network for Farmers

Over the years, by producing numerous workshops, discussion groups, and promoting initiatives in favor of local farming, the VPA has succeeded in fostering open-mindedness and an open philosophy for the promotion of the local agricultural ecosystem. In the past few years in France, tensions between conventional farmers and organic farmers have increased [25,74,75]. However, in the Versailles Plain, these tensions were not observed during fieldwork. All interviewed farmers asserted that they do not want to oppose stakeholders and they work with everyone to go "in the same direction". All together, they claim to participate in the same goal of protecting and developing farming in the Versailles plain. An open-minded state and moderate position are necessary values for integration in the VPA network.

> "Goodwill is a key word that we have found for the last two years, it is in our charter, it is central. [ . . . ] It's also a generational thing: my generation wants to live together well. Here on the plain, there is a kind of tranquility and people are well aware of how lucky they are. On the plain, the goodwill is structured with the VPA in putting the actors around the table. [ . . . ] Here, there are two big unions, why such a marked cleavage? It's impossible for them to get along, it shows a problem, it shows that there are two different farming that can't get along, and the message is very negative. If I had to do politics, which I don't want to do, I would make a speech to bring the two together. There are different methods, but we are all going in the same direction for the planet and the people." Food artisan, Versailles plain

This open-mindedness and the VPA network have fostered the development of a stakeholders' microcosm who, on a local scale, all know each other and exchange easily. This network is reinforced by discussion groups of the VPA: each college of the association meets several times a year, and the association invites professionals to offer training days on biodiversity, sustainable agricultural practices, and diversification. In 2019 for example (before the COVID-19 pandemic), the VPA organized: three meetings of each college, two local fairs around farming activities in the plain, five sciences meetings around environmental and farming issues, three exhibitions, two guides on local biodiversity, and two public events. These activities allowed the farming sector's stakeholders, from farmers to food artisans, to interact and setup joint projects. Thus, the VPA has been able to intervene actively in the establishment of new farmers (new entrants into farming and successors), but also in the development of the processing activity of agricultural products. These are indeed two key moments in peri-urban agriculture with very specific challenges.

The VPA helps new entrants into farming and successors to setup in the plain by disseminating takeover opportunities for farmland and farm buildings. It assists farmers with funding applications and helps to integrate local food chains by bringing together young farmers who wish to establish themselves. It also assists local NGOs or political

actors who seek to develop a short-food supply. Thereby, new farmers (new entrants or successors) who wish to setup in the plain can quickly get information and contacts of key stakeholders to facilitate their integration in the local farming sector. Since 2012, several new entrants into agriculture and food artisans have been established with the help of the VPA (see Figure 3). These latest setups in the plain have led to the regeneration of food production and sales systems on a territory scale. New generations of farmers are setting-up on little plots (2–5 ha) and they are integrating their activities in local food-supply chains by diversifying their activities as much as possible: market gardening activity associated with livestock; sale of transformed products at the farm or in local shops and markets. Market gardeners have very diverse products, as they sell their vegetables directly to inhabitants in the form of baskets through Associations for the Maintenance of Peasant Agriculture (AMAPs in French).

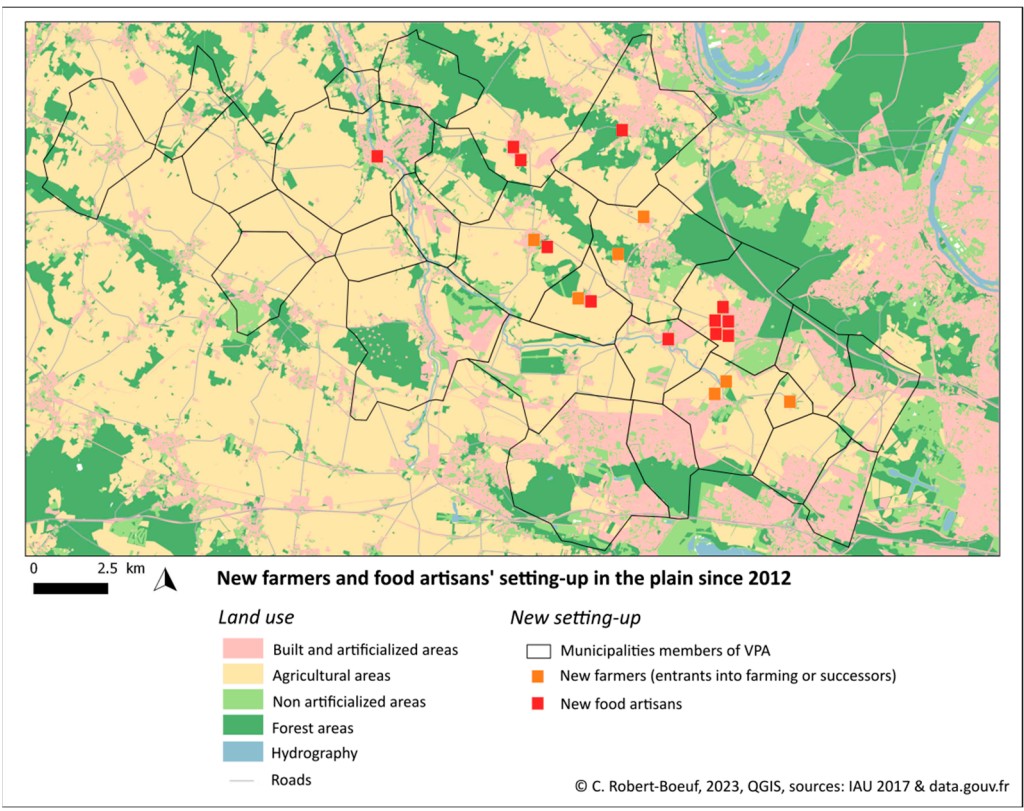

**Figure 3.** New setting-up of farmers and artisans in the Versailles plain thanks to VPA.

Moreover, in 2017, the association supported the creation of the Cooperative for the Use of Agricultural Materials (CUMA in French) in the Versailles plain, which brings together six vegetable farms. The CUMA allows farmers to invest in new quality equipment.

The VPA does not only help farmers, but also aims to develop an economic microcosm around farmers to promote local economic sectors. From this perspective, the role of local food artisans is important. There are few local artisans in the Île-de-France Region because suitable premises (with strict sanitary standards) are rare or very expensive. In addition, the demand for processing activities is also increasing from farmers, who increasingly want to be able to process their own products, but do not always have the financial means to do so. To meet this demand, the VPA is working to link farmers and food artisans to create collaborative projects. For example, in the Saint-Nom-La-Bretèche municipality, the VPA succeeded in setting-up a food artisan on a farm where the farmer had free premises and could host food-processing activities. The VPA helped provide political support and put the farmer and food craftsman in contact. The craftsman then setup his business to produce his own food products (for which the raw material was taken in the plain), but also offered

local farmers the use of his premises for their own processing needs. The setting-up project not only allowed the opening of a business, but also created a local synergy that met a local need.

In the network, the VPA has a core role in the relationship with the local political world. The facilitation and lobbying work of the VPA favors political support for farming projects. The VPA helps mayors to create local food supply chains by connecting interested farmers and elected officials and by training local elected officials on how to integrate agriculture into urban planning or how to use local food policies for sustainable farming projects.

> *"I have benefited so much from the VPA, it's huge. It is a unique association because it supports all aspects of the Plain: heritage, culture, agriculture. It manages the whole area beyond the administrative division of the different agglomerations. The VPA knows the area well and is identified by the farmers as a real stakeholder who provides resources without necessarily being political, so that gives confidence for farmers. [ . . . ] At the municipality level, we wanted to produce a local agriculture guide with a profile of each farmer, and I contacted the VPA because they have all the contacts. I couldn't have done it without the VPA. In our projects, the VPA helps us, I call on them very regularly."* Elected official, Versailles plain.

Previous cases study analyses highlighted the difficulty for local food systems to connect production and distribution [38,73]. The VPA case study shows that local governance can successfully link food production and food distribution at local the level, integrating various food stakeholders in the plain.

This governance proved to be effective during the COVID-19 pandemic, as in spring 2020, farmers and food artisans opened 40 sales outlets in the plain, almost doubling the number of outlets compared to before the health crisis. Here, we agree with recent research on the effectiveness of local food systems in the face of crises, and we reaffirm the importance of these types of governance for food systems' higher resilience [76–78].

*3.3. Transition to a More Sustainable Farming and Reassessment of Rurality through Territorial Identity*

Actions of VPA since 2004 have produced a strong territorial identity [50,79] through the agricultural and geographical landscape of the plain. Indeed, stakeholders' feelings of belonging are linked to a particular and common representation [50] of the plain territory. The landscape is characterized by a combination of fields and wooded areas with an agricultural open landscape (cereal fields). As we have already said, the plain can easily be recognized by the agricultural plain landscape with the Ru de Gally (Gally's rivulet) in the center and the wooded hillsides in the northern and southern plateaus. This view is central and produces a common territory recognized by all stakeholders: the plain is a common space that all stakeholders must protect and maintain. Besides, this landscape is constructed on a strong historical heritage: the plain is in the continuity of the Versailles Castle's park and it was originally maintained at the request of Louis XIV, who wished to keep the view and the landscape perspective from his gardens. Up until today, 2600 ha of farmland in the east of the plain are classified to protect this landscape and view from Versailles Castle. On these farmlands, all building construction is prohibited, and farmers must apply for derogation when they wish to develop new infrastructure. These historical, geographical, and landscape features produced the territorial identity of the plain, where farming activities are at the core of the Versailles Plain identity and are maintaining its geographical coherence. This plain landscape and identity were preserved by the VPA thanks to the writing of a landscape charter in 2014, which was signed by all the mayors of the plain. This charter affirmed the agricultural, economic, and landscape aspects of the plain and reinforced the place of farmers as central stakeholders.

Now, the territorial identity of the plain is really appropriated by all stakeholders. All the interviewees entertain a very positive representation of the plain's environment and its agricultural character: they claim to work and/or live in a territory rich in terms of

biodiversity (farmers and local NGOs are aware of the flora and fauna of the plain) and consider that they are lucky to be able to have settled there.

*"The sellers are good, people are demanding, they want local products and in the plain, there is a lot of productions, the diversity is incredible, you can buy all products locally. You can get everything. [ . . . ] We are in the fields, but ten minutes away we are in a big city, it's special. And that's a huge advantage: my business, at the beginning I was in Nancy [East of France], I didn't think I could sell my products there, the buying power is much lower, there are a less people, the land is not the same. My business can work because I'm here in this plain, it was clearly part of my business plan. It's more difficult to set up here because everything is more expensive, and I'm very lucky to have my family's land, but if I had set up somewhere else, I would have had to lower my prices and everything, I wouldn't have been able to do even my training courses, because I wouldn't have had enough customers around." Successor and new farmer, Versailles plain.*

The territorial identity of the Versailles Plain is based on an agricultural and rural landscape; it is opposed to the surrounding cities by the territorial grid with small villages and a lower density. The rural features of the territory thus become an element of distinction and appropriation, which makes it possible to oppose the urban sprawl and the densification of built-up areas in the plain's municipalities. In the peri-urban context of the Versailles Plain, these features give the opportunity to claim a new way of living and an alternative territorial development, refusing the negative aspects of urbanization. Here, we agree with Itçaina's studies on Basque territorial identity as a way of preserving and developing local food production [80]. We confirm that the construction of an identity around a rural and agricultural territory is an important springboard for the mobilization of local stakeholders and farming preservation (even if the Basque example shows a more complete form of institutionalization). This territorial identity is now slowly changing under the impulse of civil society, and new farmers and food artisans to move towards a more local and sustainable agriculture. In fact, most new farmers in the plain are market gardeners, stockbreeders, or beekeepers who have setup on small plots and introduce a break with the traditional large cereal farms. They have produced new landscapes with vegetable fields and greenhouses. They promote a more environmental perception of the landscape with a concern for the preservation of biodiversity and a strong will to raise awareness for wildlife and plant protection. These new farmers have a common philosophy about the environment that underlines the environmental impact of agriculture and supports organic and sustainable agriculture. Those who are in conventional agriculture are also greatly reducing the use of chemical inputs and are planting hedges. More generally, farmers are integrating the preservation of biodiversity into their practices. This evolution is supported by the VPA, which regularly organizes workshops about organic or more natural farming practices. The association has also conducted several territorial diagnoses and fauna and flora surveys to monitor their evolution. More broadly, the VPA has reinforced the dialogue between environmental associations, hunter associations, local authorities, and inhabitants to reduce the impact of human activities on natural and agricultural areas.

Environmental issues were accentuated during the lockdown of the COVID-19 pandemic. The inhabitants of the large neighboring cities were restrained in a small perimeter around their homes and, therefore, began to visit the plain for leisure. The sudden increase in the number of visitors to the plain led to the degradation of some natural areas and fields and caused a lot of concern for farmers who felt "invaded". In this context, the VPA was central in establishing a dialogue between the different stakeholders by organizing workshops and meetings on this topic. After that, local authorities decided to produce awareness campaigns and recruit a field guard in the plain.

This territorial identity has, therefore, moved from a traditional cereal-based agricultural landscape to a sustainable and subsistence landscape that promotes local food supply chains and territorial balance without excessive urbanization. Therefore, according to our study, the VPA produces a transversal form of rural regeneration because it promotes new farmers' setup and an agricultural transition towards food-producing and sustainable

agriculture. More generally, it offers a positive and federating territorial identity based on the agricultural and rural character, without opposing conventional and alternative food systems, as has been analyzed in other cases [49]. This territorial identity produces positive representations of rurality, where rural and agricultural stakeholders have an active role in the construction of the plain.

Nevertheless, our case study analysis showed that the VPA is now facing new challenges. Although it has succeeded in preserving agricultural land despite land pressure and has even contributed to the setting-up of new farmers by federating the plain's stakeholders against urbanization, it presently has difficulties in bringing together people for a new territorial project. The evolution of the plain's territorial identity is not shared by everyone and reveals tensions between different generations. Tensions are first observed between young farmers (new entrants or successors) and the older ones. The new farmers want to change their practices to favor organic or more sustainable agriculture, with fewer chemical inputs and less use of tractors, which is opposed to the practices of the plain's cereal farmers. These generational tensions among farmers show that, although the VPA wants to defend lowland agriculture, it must deal with groups of farmers with different interests and different social and economic realities. This tension is consistent with the fact that few traditional cereal growers actively integrate in the VPA. To respond to these tensions, the VPA could use several levers: (i) the VPA could collaborates with more traditional farmers associations and unions that could help it to dialogue with these less invested farmers; (ii) some cereal farmers who are members of the VPA and could act as trusted intermediaries to find common projects. Yet these levers would need to be developed over time. We also noticed tensions among the inhabitants of the plain. The first generation is local and rural and knows the farming profession and its challenges. Newcomers come mostly from the city and have no knowledge of farming practices but have strong environmental claims which are not always compatible with agricultural practices (even organic). These claims tend to create new conflicts amoung farmers.

## 4. Concluding Remarks

In conclusion, our analysis shows an interesting case study of a peri-urban agricultural plain which, thanks to the VPA and its territorial governance, has succeeded in promoting rural regeneration near cities and sustainable farming. This governance is based on its capacity to be a facilitating operator and to bring together very different stakeholders from rural and urban worlds, giving them empowerment capacities at a local level to defend the plain against urbanization. The VPA organization places agricultural actors, political actors, and civil society actors on the same level, which favors exchanges and empowerment processes for farmers, who are usually excluded from decision-making procedures in peri-urban areas.

The VPA promotes economic microcosm for farmers and food artisans, which makes the agricultural and food sector's stakeholders truly active within the territory. These stakeholders produce food for the local population, landscapes, and a living environment appreciated by the inhabitants. They participate in the territorial identity and thus contribute to an evolution of the rural areas' representation and promote sustainable agriculture. The VPA case study shows an evolution towards positive regeneration processes of rural territories complemented by a reassessment of rurality and its ways of living for local stakeholders. In the Versailles Plain, agriculture is perceived as a way to create an alternative territorial development responding to urbanization processes. Preservation and development of agricultural activities and landscape in this peri-urban plain produce synergies to counter urbanization and the artificialization of land, the densification and sprawl of villages, and the destruction of natural areas. This counter process can be defined as a ruralization process because it is characterized by a change in stakeholders' rural representations, these representations becoming values and identity features for a new local network and governance.

Nevertheless, the VPA case study also shows that this kind of facilitating operator needs constant renewal of active participants as well as political and financial support on the long terms. This lack of long-term vision is still a weakness and is reinforced by the coexistence of opposed visions. Indeed, certain tensions appear between on the one hand new farmers, local populations, and local elected officials who want to turn actual farming into subsistence agriculture, and on the other hand more conventional stakeholders who wish to preserve cereal agriculture and its landscapes. Thus, VPA promotes a complementarity relationship between subsistence and cereal farming, but for the moment the association didn't find a clear and approved pathway to reach this complementarity relation.

**Supplementary Materials:** The following supporting information can be downloaded at: https://www.mdpi.com/article/10.3390/su15097219/s1, Figure S1: Interviews Guide.

**Funding:** This research was funded by the Ruralization project (GA 817642) Horizon 2020 of the European Commission.

**Institutional Review Board Statement:** The study was conducted in accordance with the Declaration of Helsinki, and approved by the Institutional Review Board (or Ethics Committee) of Human Research Ethics Committee, TU Delft on the 31 October 2019. It was also approved by the General Data Protection Regulation (GDPR) of the CNRS, in accordance with the EU GDPR.

**Informed Consent Statement:** Informed consent was obtained from all subjects involved in the study.

**Data Availability Statement:** The data are available at the CNRS office, Ladyss Laboratory, Paris Nanterre University.

**Acknowledgments:** Part of the data (online survey) was collected with the help of Hervé Brédif (Université Paris 1 Pantheon-Sorbonne), Ambroise de Montbel (AgroParisTech) and their students, in the framework of the Ruralization project. The authors also would like to thank reviewers for their comments, which helped the improvement of the paper.

**Conflicts of Interest:** The author declares no conflict of interest.

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
