# Peer review of "Promoting Rural Regeneration and Sustainable Farming near Cities Thanks to Facilitating Operators in France? The Case of the Versailles Plain’s Association Governance Model"

_sustainability, doi:10.3390/su15097219_

Round 1

Reviewer 1 Report

I.                    General conclusions:

Congratulations for the paper! Interesting, important, and actually topic! The question is if give the paper content the answer scientifically argued to the title, or try to promote the project?

Please, try to synchronize the title, aims in abstract and introduction, method, results, and discussions!

Please, pay attention to abbreviation (must be defined before the use).

 II.                  Particularly observations

1.       Introduction

Generally, analyze is clear, comprehensive and relevance to the field.

In this area the presented paper’s aim can be more synchronized with title and other chapters.

 2.       Material and methods

Please keep in mind the results are reproducible based on the details given in the methodology section. The methods can be presented more clear, re-structured on technical logical flow. The introduction is focused on ruralization and cap 2.1 insist on rural regeneration as disunited concepts. Also, seems to be justification more than theoretical framework.

For readers is important to previously explain the significance of UAA or CAP.

In text some variables from table 1 can be detailed (Elected officials and institutions or Others), and justified for better understanding.

 3.       Results

At results must be presented own work. Some parts must be moved at methodology. If some results are compared with those from other authors, it must change the title in „Results and discussions”.

Between text from rows 173-175 and 177-180, seems to be misunderstanding. Also Table 2 and rows 191-193, etc.  

The conclusions (rows 389-398), must be compared wit statistic in table 2.

 4.       Discussions and concluding remarks

Usually at part of discussions own results are compared with others (references). Conclusions must be synchronized with title and aim.

Author Response

Thank you for your review, it helped me to improve the paper. Here are the changes I made following your comments. I hope it will meet your expectations.

  1. Aim of the paper and conclusion were synchronized with title
  2. Methodology has been improved to be more clair and precise and theoretical framework reorganized.
  3. Results and discussions have been gathered in the same part of the paper. Unclear part of the text have been modified.
  4. Conclusion has been synchronized with title.

Reviewer 2 Report

The article present a critical issue and is well documented and written. The arguments are clearly stated and the data clearly presented untill the discussion that needs to be rewritten as the discussion presents new elements "However, our case study analysis showed that the VPA is now facing new challenges" that should have been presented in the results part.

The discussion, different from the conclusion, must respond to what the results mean. You need to make the Discussion corresponding to the Results, but do not reiterate the results. You need to compare the published results by your colleagues with yours (using some of the references included in the Introduction). This part must thus be rewritten in order for the article to be published.

We also advise some editing to the article.

Author Response

Thank you for your review, it helped me to improve the paper. Here are the changes I made following your comments. I hope it will meet your expectations.

  1. Results and discussion have been put in the same part of the paper, apart from the conclusion.
  2. The results have been compared to other published studies.

Reviewer 3 Report

L’article est concis et bien écrit, agréable à lire. Après lecture de l’article, je comprends que l’organisation étudiée opère comme un facilitateur et que, à la différence de ce qui a été montré précédemment, les agriculteurs sont moteurs et centraux dans l’organisation.

Toutefois, il me semble que certains maillons de l’argumentaire manquent et voici quelques suggestions : mieux situer les enjeux scientifiques et opérationnels de la ruralisation (et en lien du rural / urbain et du périurbain), ceux d’une gouvernance équitable et de l’empowerment des agriculteurs dans les systèmes agrialimentaires territorialisést ; ne pas limiter les résultats à ce qui sert directement la démonstration mais considérer aussi ce qui permet de nuancer l’hypothèse ensuite en discussion. Ci-dessous les remarques qui ont émergé au fil de la lecture. Merci pour cet article stimulant.        

-          Sur le plan conceptuel, un certain nombre de notions ou cadres sont mobilisés mais certains arguments reposent sur des implicites qu’il faudrait mettre au jour.

o   Les notions de rural et d’urbain mériteraient d’être explicitées : s’il ne s’agit pas nécessairement de les définir – on sait combien cette démarche peut être aventureuse – je pense qu’il serait de bon ton de proposer des caractéristiques, ou principes, qui permettent de comprendre comment ces termes vont être mobilisés dans la suite de l’article.

o   De même, on aurait besoin de comprendre quels sont les enjeux liés à la ruralisation et quels espaces sont en fait concernés (est-t-on sur la ruralisation d’espaces urbains ou bien les espaces ruraux peuvent être eux-mêmes concernés ? je me pose cette question dans un contexte où la ruralité peut ne pas être bien vécue, dans les ajustements en termes de mobilité, de sociabilité et même de rapport à la nature qu’ils peuvent occasionner). La ruralisation est présentée ici comme le contre-pied de l’urbanisation et comme quelque chose de foncièrement positif, mais j’ai besoin qu’on me convainque, qu’on me le démontre.

o   Aussi, si la ruralisation n’est pas forcément définie en introduction, étant donné que dans le papier « we wish to provide some elements of definition and evaluate the potential of this process in France » (ligne 35) et on en a d’ailleurs une réponse en fin de conclusion (ligne 435), ce serait intéressant de présenter les grandes caractéristiques de l’urbanisation, ou peut-être ses effets pervers en intro.  

o   Le passage du rural à l’agricole est un peu rapide (ligne 36), alors même que la relation entre agriculture et rural est bien documenté, discuté. Il me semble qu’il serait intéressant de mieux assurer la transition de l’un à l’autre, et peut-être aussi d’interroger le rôle de l’évolution de l’engagement citoyen dans leur alimentation, de l’évolution des politiques publiques de l’agricole à l’alimentaire comme des facteurs impactant les formes de gouvernance dans les territoires et la place des agriculteurs dedans. Ça permettrait, il me semble, de mieux mettre en avant l’originalité de votre hypothèse, votre parti-pris.

o   La question des relations entre acteurs (l61-63) et la notion de gouvernance sont centrales dans votre article, mais vous ne précisez pas votre positionnement vis-à-vis de la littérature (ou pas assez). Il me semble d’ailleurs qu’il serait pertinent de mieux expliciter le rôle de l’urbanisation, de la péri-urbanisation dans l’évolution des relations entre acteurs, et peut-être dans l’expression de différentes formes de gouvernances (ou dans la place occupée par diverses catégories d’acteurs, et notamment les agriculteurs).

Dans le même ordre d’idée, la notion d’empowerment (l99) est mentionnée. Après lecture de l’article, je comprends que c’est l’empowerment des agriculteurs du territoire (mais peut-être que je me trompe ?). Ce serait peut-être intéressant de donner une plus grande place à cette notion dans votre problématisation ?

o   C’est difficile de comprendre la place du péri-urbain entre rural et urbain. Peut-être qu’une simple phrase ou deux à ce sujet permettrait d’ajouter de la clarté ?

o   VPA est présenté dès le début comme un « facilitating operator » (ligne 65). est-ce qu’il s’agit d’une notion, auquel cas merci de vous référer à l’état de l’art ? ou alors est-ce que ce n’est pas déjà la réponse dans la question i.e. votre but est de montrer que VPA est un facilitating operator et donc qu’il contribue à une meilleure gouvernance locale ? Suivant la réponse à ces questions, veiller à mieux formuler certaines phrases.

o   D’autres notions comme « l’identité territoriale » ou les « valeurs » mériteraient enfin d’être explicitées ou référées à de la bibliographie pour mieux en comprendre l’usage que vous en faites.

-          Sur le plan méthodologique

o   L95 : en quoi le VPA est un cas d’étude intéressant, original, pertinent compte tenu de votre problématique ?

o   L’enquête est dite collaborative : en dehors de la bonne volonté des acteurs à répondre aux questions des étudiants et chercheurs, quel était l’intérêt des acteurs à y prendre part ? faisaient-ils face à un problème particulier qu’ils souhaitaient voir travaillé ? Quelles perspectives ont été tirées de cette recherche ?

o   Il serait intéressant de développer le contenu de l’enquête : qu’est-ce qui était demandé dans les questionnaires ? quelles thématiques abordées pendant les entretiens semi-directifs ou pendant les focus group sur la ruralisation ?

-          Sur la présentation du cas d’étude

o   Si la dimension agricole est bien présentée, il manque une présentation plus générale du territoire : la population (et peut-être même la part des agriculteurs dans la population des communes concernées), la place d’autres espaces, notamment la forêt qui est significative… Les axes de transport pourraient aussi être présentés sur la carte 1, pour peut-être mieux mettre en avant le caractère Péri-urbain, banlieusard d’un tel territoire. Cela permettrait peut-être de mieux situer les enjeux liés à une revitalisation/régénération rurale d’un tel territoire.

-          Sur la présentation des résultats et leur discussion

o   L166-171 : est-on sur du teasing ? est-ce que c’est quelque chose qui va être démontré ou bien c’est posé comme un fait ? Il me semble qu’on est déjà sur un paragraphe conclusif et que sa place n’est pas forcément là (ajoute de la confusion)

o   Il serait peut-être intéressant d’avoir quelques éléments historiques concernant l’association, notamment concernant les acteurs en présence et les modes de gouvernance. Est-ce qu’il s’agit du mode de gouvernance originel ? Qu’est-ce qui fait qu’il a été pensé comme ça ?

o   L361-363 : la question rurale réapparait, ici comme un principe fondateur de l’identité territoriale. Mais je ne sais toujours pas de quoi elle est faite. J’ai besoin de plus d’éléments à ce sujet. Comment les acteurs expriment-ils leur ruralité ?

o   L400 – 421 : je trouve dommage que ces éléments de discussion, qui sont finalement très factuels et nuancent l’affirmation comme quoi le VPA est un « facilitating operators » apparaissent ici. ce sont, à mon sens, des résultats qui mériteraient d’être présentés comme tel, peut-être en 3.2. ou en 3.4 (sur sa capacité à promouvoir des modes de production plus respectueux de l’environnement et dédiés à l’alimentation de la population). Si l’on reprend le cadre proposé par Lamine (qu’elle développe aussi dans d’autres article comme celui dans Sociologie Ruralis (2015 - Sustainability and Resilience in Agrifood systems: reconnecting Agriculture, Food and the Environment), ce que vous présentez sont en fait des éléments d’une controverse sur la transition agroécologique.

Il me semble que ce genre de développement permettrait aussi de mieux expliciter ce « paradigmatic change » que vous évoquez en fin de conclusion, mais qui n’est pas totalement illustré dans les résultats (ou mentionné comme tel, il me semble). Celui-ci ne se fait pas sans tension ou sans déperdition, ce qui mérite d’être mis en valeur.

Il serait par contre intéressant de développer la mention suivante : « difficulty today in bringing together people for a new territorial project » : quel projet territorial ? quels éléments d’explication à ces difficultés ?

En bref, nuancer vos propos ne signifie pas infirmer votre hypothèse. Au contraire, il me semble que ça peut lui donner de la robustesse.

Author Response

Thank you for your review, it helped me to improve the paper. Here are the changes I made following your comments. I hope it will meet your expectations.

  1. Concerning conceptual work: I introduced more the concept of urbanization in order to explain differences between rural, peri-urban and urban areas, as you suggested. To do so, I reorganised the introduction part accordingly.
  2. I add some bibliographic references about empowerment, territorial identity and values. The empowerment concept has been integrated more in the introduction part of the paper
  3. I have modified the presentation of the VPA case study, with more context (data about local population sociology, local agricultural sector and land use)
  4. I put the interview guide in supplementary materials (S1) and I have detailed the collaborative part of the research better
  5. I modified the map to show the roads as requested
  6. I modified the results and confronted them to published papers in order to integrate the discussion in the results part.
  7. The conclusion has been modified.

Author Response

Thank you for your review, it helped me to improve the paper. Here are the changes I made following your comments. I hope it will meet your expectations.

  1. I modified keyword list
  2. I modified misspelling ("etc.") in the text.
  3. I added international scientific litterature in the introduction part (especially about food system models and urban-rural cooperation models)
  4. I added statistical data on local agricultural context
  5. My work is based on a qualitative research with mainly qualitative data. I did add some data about agricultural sector in the region, as you suggested.
  6. As you suggested, I added interviews guide in supplementary materials (S1)
  7. I have confronted more published litterature with my own results in order to do a real discussion part (discussion part has been integrated in the results part)
  8. I must emphasise that my work is based on a qualitative research that falls within the stream of social geography and social sciences in general. Therefore, the study is based on interviews, not on macroeconomic statistical analysis. This explains the presence of mainly qualitative data. However, your feedback has allowed me to reflect on the construction of my argument and I have been able to improve it by clarifying certain data and arguments:
  9. I have detailed more activities of market gardeners in the plain.
  10. I have added specific data on the activities of the VPA (l352-355), I have added statistical data on the agricultural context of the plain in the case study's presentation part.

Reviewer 5 Report

Find my recommendations in the attached file.

Author Response

Thank you for your review, it helped me to improve the paper. Here are the changes I made following your comments. I hope it will meet your expectations.

  1. I reorganised the methodology part of the paper
  2. I put the discussion part in the results part, in order to separate discussion and conclusion
  3. More recent bibliographic references have been added

Round 2

Reviewer 2 Report

thank you for having taken into account my remarks. There is still a need of minor editing: spelling mistake line 518 but there might be more

Author Response

Thank you for your review.

I have corrected several spelling mistakes, as requested.

Reviewer 3 Report

Thank you for your modifications. Here are some secondary remarks.

l108-111, could you reformulate your hypothesis ? the first part shows limit of previous studies identifying types of stakeholders, while your hypothesis is about a process. Do you want to show that all stakeholders are involved ? in that case, please write it so.

The presentation of your case study is clearer and easy to read.

Some of your results makes me think about studies that were realised about an alternative agricultural extension service (EHLG - Pays Basque). Maybe can you check itçaina's papers, about agricultural institutional representation, to strengthen or discuss some of your statements ? or even to explore comparative research on a longer term.

The rural(ization), urban(ization) notions are more clearly employed in the paper. Your contribution about ruralization is clearer too.

In the end of your results, you could develop a little bit more about stakes and difficulties of inclusion while some stakeholders tend to retire from the VPA or avoid it (cereal farmers). This is an important element when dealing about governance : how good can it be if some stakeholders are not represented ?  Are the levers that the VPA could use to integrate them ?

Your conclusion is concise and clear.

It was a pleasure to read your paper.

Author Response

Thank you for your review.

  1. I reformulated my hypothesis.
  2. Thank you for this new bibliographic reference, it was very interesting for the discussion of some of my results. I integrated the Itçaina work in the results part of the paper.
  3. I completed arguments on VPA difficulties and levers to integrate cereal farmers in the local governance (l521-525).

Reviewer 4 Report

Dear Author,

Thank you very much for the corrections made. Generally, I have no comments. However, I'm wondering about the poll you've attached. There are given issues, however, there is no information about what type of questions they were, whether they were open or closed. This is quite important information from the results analysis perspective. Please consider posting a sample survey with responses. This would greatly facilitate orientation in the type of collected data.

Kind regards

Author Response

Thank you for this new review.

I explained more the interviews process in the beginning of the Figure S1. I hope it  meets your expectations.

As all the interviews were conducted in French and stakeholders are different (farmers, elected official, ...), it can be complicated to post a complete and representative sample of survey with responses.